# Neurovascular Cell Death and Therapeutic Strategies for Diabetic Retinopathy

**DOI:** 10.3390/ijms241612919

**Published:** 2023-08-18

**Authors:** Toshiyuki Oshitari

**Affiliations:** 1Department of Ophthalmology and Visual Science, Chiba University Graduate School of Medicine, Inohana 1-8-1, Chuo-ku, Chiba 260-8670, Japan; tarii@aol.com; Tel.: +81-43-226-2124; Fax: +81-43-224-4162; 2Department of Ophthalmology, School of Medicine, International University of Health and Welfare, 4-3 Kozunomori, Narita 286-8686, Japan

**Keywords:** diabetic retinopathy, neurovascular unit, neurovascular cell death, apoptosis, necroptosis, ferroptosis, pyroptosis, neuroprotection, vasoprotection

## Abstract

Diabetic retinopathy (DR) is a major complication of diabetes and a leading cause of blindness worldwide. DR was recently defined as a neurovascular disease associated with tissue-specific neurovascular impairment of the retina in patients with diabetes. Neurovascular cell death is the main cause of neurovascular impairment in DR. Thus, neurovascular cell protection is a potential therapy for preventing the progression of DR. Growing evidence indicates that a variety of cell death pathways, such as apoptosis, necroptosis, ferroptosis, and pyroptosis, are associated with neurovascular cell death in DR. These forms of regulated cell death may serve as therapeutic targets for ameliorating the pathogenesis of DR. This review focuses on these cell death mechanisms and describes potential therapies for the treatment of DR that protect against neurovascular cell death.

## 1. Introduction

According to the International Diabetes Federation Diabetes Atlas, the global prevalence of diabetes in individuals aged 20–79 years old was approximately 10.5% (540 million people) in 2021, which will increase to 12.2% (approximately 780 million) in 2045 [1]. Diabetic retinopathy (DR), defined as the tissue-specific neurovascular impairment of the interdependence between cells comprising the neurovascular unit, is a major complication in patients with type 1 and type 2 diabetes [2]. A recent meta-analysis indicated that the global prevalence of DR was 22.7%, that of vision-threatening DR was 6.17%, and that of clinically significant macular edema was 4.07% [3]. In this study, the estimated number of patients with DR by 2045 was updated. The number of patients with DR was 103.12 million in 2020 worldwide, and this number is expected to increase to 160.50 million by 2045 [3].

The neurovascular unit consists of multiple cells, including retinal ganglion cells (RGCs), bipolar cells, amacrine cells, horizontal cells, Müller cells, astrocytes, microglia, endothelial cells, and pericytes [4,5,6]. The interdependence of these cells is essential for maintaining a healthy retinal environment, and the impairment of this interdependence under chronic hyperglycemia triggers the development of DR. Neuronal abnormalities, including neuronal cell death, are irreversible changes thought to precede vascular abnormalities in the early stages of DR [7,8,9,10]. Several previous studies have indicated that more retinal neurons in the ganglion cell layer, including RGCs, die by apoptosis in human diabetic retinas than in retinas without diabetes [11,12,13]. Thus, neuroprotective and regenerative therapies should be considered as early interventions to prevent vision loss in patients with DR.

Pericytes are command centers for maintaining the homeostasis of retinal vessels, including the formation of the blood–retinal barrier [14,15]. For example, pericytes regulate the expression of the vascular endothelial growth factor (VEGF) receptor 2 and angiopoietin-2 via the forkhead box protein O1 pathway, followed by the regulation of VEGFA signaling [14]. Pericyte loss may cause microaneurysm formation, which occurs prior to endothelial cell loss in experimental DR [16]. Previous studies have indicated that pericyte apoptosis is higher in human retinal samples from patients with diabetes than in retinas from patients without diabetes [17,18]. Thus, the prevention of pericyte loss is a therapeutic option in the early stages of DR.

Endothelial cells are components of the blood–retinal barrier. Endothelial cells connect horizontally with tight junction proteins, including occludin, claudin-5, and zonula occludins-1, and the expression of these tight junction proteins is reduced in DR, resulting in vascular leakage [19]. Previous studies have indicated that endothelial cell apoptosis occurs in experimental DR [20] and human diabetic retinas [21]. Endothelial cell loss leads to the formation of acellular capillaries in DR. Because cellular capillaries have no cellular function, vessels can easily collapse owing to fluctuations in blood pressure and accelerate vascular leakage because of the lack of barrier function. Tien et al. demonstrated that the gap junction protein connexin 43 expression was significantly decreased in human diabetic retinas compared to that in non-diabetic retinas, and that the decrease in the number of connexin 43 plaques was parallel to the pericyte loss and acellular capillaries [22]. These results indicate that disturbances in cell–cell communication are related to the development of vascular cell death in human DR. Gap junctions are also associated with neuronal cell death and may serve as therapeutic targets for DR [23].

Over a decade ago, cell death was classified into two main patterns: apoptosis and non-apoptosis (or necrosis). Apoptosis is strictly regulated by the intrinsic and extrinsic cell death pathways. Apoptotic cells die proactively without inducing inflammation. In contrast, necrosis is thought to be passive cell death with no regulation; thus, necrotic cells are ruptured and distribute various toxic substances, including enzymes and nucleotides, resulting in the induction of tissue inflammation surrounding necrotic cells. Most retinal cell death in DR has been identified as apoptosis, which can be detected by terminal deoxynucleotidyl transferase dUTP nick-end labelling (TUNEL) staining [24]. However, various necrosis-like cell death pathways, such as necroptosis, ferroptosis, and pyroptosis, have recently been identified in DR [25]. In this review, we focus on various types of retinal cell death and update the literature on the novel regulation of cell death in DR. Possible therapeutic approaches for treating DR are described.

## 2. Various Types of Retinal Cell Death in DR

Over the last two decades, apoptotic cell death has been observed in the retinas of patients with diabetes [11,12,13,17,18,21,22]. A previous human diabetic retinal study indicated that most degenerating neurons show activated caspase-3 immunopositivity; thus, most degenerating neurons appear to die by apoptosis [12]. Apoptosis is a strictly regulated cell death (RCD) event that includes chromatin condensation, DNA fragmentation, and the formation of small apoptotic bodies, which results in phagocytosis by the surrounding cells without inducing an inflammatory reaction. Because apoptosis is believed to be a major form of cell death, the first topic of retinal cell death in this section is apoptosis.

### 2.1. Apoptosis in DR

Apoptotic cell death occurs in various types of retinal cells, such as pericytes [17,18], endothelial cells [20,21], and neuronal cells [7,8,9,10,11,12,13], and is associated with the pathogenesis of DR [5,6,24]. Neuronal cell death is an irreversible change directly related to vision loss in patients [12,13]. As neuronal cell death occurs in the early stages of diabetes, early intervention, including neuroprotective therapies, is required to sustain visual function in patients with DR. The elucidation of the precise mechanism of neuronal cell death in DR is urgently required to establish neuroprotective therapies. However, the precise mechanisms underlying neuronal cell death in DR remain unclear. A possible mechanism of apoptotic cell death in DR is shown in Figure 1. Apoptotic cell death pathways are broadly divided into two pathways: the intrinsic pathway, which is activated during development, DNA damage, or chemical injuries, and the extrinsic pathway, which is activated via death receptor signals [26,27]. In the intrinsic pathway, a sensor protein, c-Fos/c-Jun (activator protein-1 (AP-1)), transfers cell death signals to the mitochondria [28,29], resulting in the activation of caspase-9 and -3 in cultured retinas [30,31] and human diabetic retinas [12,13]. In the extrinsic pathway, tumor necrosis factor-α (TNF-α) and TNF receptor 1 (TNFR1) are associated with retinal neuronal cell apoptosis [32], retinal pigment epithelium apoptosis [33], and retinal endothelial cell apoptosis [34] under diabetic stress. However, in neuronal cells, the extrinsic pathway is thought to induce the activation of the intrinsic pathway by translocating truncated Bid (t-Bid) to the mitochondrial membrane after cleavage by caspase-8 (Figure 1) [35]. Most researchers have indicated that endoplasmic reticulum (ER) stress is associated with the pathogenesis of DR [29,36,37,38,39,40]. Briefly, ER stress sensors include the inositol-requiring ER-to-nucleus signaling protein 1 (IRE1), protein kinase-like ER eukaryotic initiation factor 2-alpha kinase (PERK), activating transcription factor-6 (ATF6), and inositol trisphosphate receptor (IP3R) (Figure 1). Activated PERK phosphorylates eukaryotic initiation factor-2α (eIF-2α), resulting in the increased expression of activating transcription factor 4 (ATF4) [41]. The persistent activation of the PERK-ATF4 pathway facilitates apoptosis by inducing the transcription of CCAAT/enhancer-binding protein homologous protein (CHOP). CHOP induces the expression of Bcl-2 interacting mediator of cell death (BIM) and induces apoptosis by activating Bax/Bak and inhibiting Bcl-2 [42].

Activated IRE1 recruits TNFR-associated factor 2 (TRAF2) followed by activating apoptosis signal-regulating kinase 1 (ASK1) and c-Jun-N-terminal protein kinase (JNK) [43,44]. Previous studies, including ours, indicated that JNK is critically associated with ER stress-induced retinal cell death under diabetic stress [13,29,40,45,46,47]. Xu et al. indicated that the anti-apoptotic effect of melatonin is associated with the suppression of the ATF6-CHOP pathway in the brain [48]. In the diabetic rat retina, ER stress markers, including ATF6 and CHOP, are upregulated [49,50], and vitamin B12 supplementation prevents photoreceptor cell death by suppressing ER markers [50]. Taken together, these results indicate that the ATF6-CHOP pathway is involved in retinal cell death in diabetic retinopathy. Under excessive ER stress, Ca^2+^ is released from the ER via IP3R, which induces mitochondrial Ca^2+^ accumulation [51]. Sustained Ca^2+^ accumulation in mitochondria promotes the mitochondrial permeability transition, followed by the release of cytochrome c and apoptosis-inducing factor (AIF) (Figure 1) [51]. A previous study indicated that IP3R-related Ca^2+^ release is partly associated with capillary degeneration in DR [52]. Under normal conditions, phosphatidylserine (PS) is distributed in the intracellular phospholipid bilayer via flippases. In contrast, the scramblase exposed the PS to the extracellular layer of the bilayer. In previous studies, adenosine triphosphatase type 11C (ATP11C) and ATP11A, which belong to the type IV P-type ATPase family, were identified as ubiquitously expressed flippases in the cell membrane [53,54], and Xk-related protein 8 (XKR8), which belongs to the XKR family, was identified as a scramblase in the cell membrane [55]. During the late phase of apoptosis, the active form of caspase-3 cleaves flippases and scramblases, resulting in their flipping off and scrambling, respectively. As a result, PS was exposed on the surface of the cell membrane from the inside of the bilayer. The exposure of PS on the surface of the membrane bilayer is reflected as “Eat me” or “Find me” signals for phagocytosing cells, such as macrophages (Figure 1). In patients with diabetes, PS is more exposed on the membrane of erythrocytes than in healthy individuals via the inhibition of flippase-like activity by tubulin [56]. However, there are no reports on the use of flippases and scramblases in DR. Further studies are required to elucidate the association between flippases and scramblases and the pathogenesis of DR.

### 2.2. Pyroptosis in DR

The Nomenclature Committee on Cell Death (NCCD) defines pyroptosis as RCD accompanied by the formation of plasma membrane pores by the gasdermin protein family, which is often induced by inflammatory caspase activation [57]. Diabetes mellitus is a chronic inflammatory disease; thus, pyroptosis, an inflammation-related RCD, is associated with the pathogenesis of diabetes mellitus and DR [58,59]. A hypothetical mechanism of pyroptosis is shown in Figure 2.

The first priming signals of the classical pathway of pyroptosis are to bind pathogens or cytokines, including TNF-α and IL-1β, to Toll-like receptor (TLR) followed by activating NF-κB [61] (Figure 2). NF-κB transcripts include pro-IL-1β, pro-IL-18, and a component of the inflammasome, NLRP3 (Figure 2). The induction of TLR4 expression in retinal endothelial cells has been observed under high-glucose conditions [62]. In addition to TLR4, TLR2, NF-κB, TNF-α, and IL-8 are increased in RGCs under high glucose exposure [63]. The second signal is exposure to NLRP3 agonists, which include damage-associated molecular patterns (DAMPs) and pathogen-associated molecular patterns (PAMPs). DAMPs and PAMPs induce mitochondrial damage, followed by increased reactive oxygen species (ROS) production and NLRP3 activation [64,65] (Figure 2). The activated NLRP3 undergoes oligomerization, resulting in the recruitment of ASC, MEK7, and pro-caspase-1, followed by the formation of an active NLRP3 inflammasome [66] (Figure 2). The activated NLRP3 inflammasome activates caspase-1 by cleaving pro-caspase-1 and produces mature IL-1β and IL-18. Furthermore, caspase-1’s cleavage of gasdermin D (GSDMD) and the 33-mer N-terminus of GSDMD results in GSDMD pores approximately 22 nm in diameter in the plasma membrane [67,68]. The GSDMD pores release low-molecular-weight DAMPs, IL-1β, and IL-18 to extracellular spaces (Figure 2). Furthermore, the passive plasma membrane rupture mediated by NINJ1 exacerbates inflammatory reactions by releasing high-molecular-weight DAMPs [60]. In the non-classical pathway, endotoxins such as lipopolysaccharide activate caspase-4/5/11’s cleavage of GSDMD in a caspase-1-independent manner [69]. Several studies have indicated that NLRP3 activation, caspase-1 activation, and the upregulation of IL-1β and IL-18 are found in retinal endothelial cells in vitro and in vivo [70,71,72]. Several previous studies using human retinal pericytes have indicated that GSDMD activation and pore formation followed by releasing IL-1β and IL-18 were induced by high glucose exposure in a dose- and time-dependent manner [73] and that in human retinal pericytes exposed to advanced glycation end-products, caspase-1 and GSDMD were activated followed by increases in IL-1β, IL-18, and lactate dehydrogenase (LDH) [74]. These results indicate that pyroptosis is partly associated with pericyte loss in DR. In Müller cells, angiotensin-converting enzyme, the active form of caspase-1, and IL-1β were increased under diabetic stress in vitro and in vivo, and the NLRP3 inhibitor MCC950 reduced their expression [75]. These results indicate that the NLRP3 inflammasome pathway is activated in Müller cells in DR. A recent study indicated that the knockdown of transient receptor potential channel 6 reduced pyroptosis in rat retinal Müller cells by inhibiting ROS and NLRP3 [76]. A previous study indicated that LDH release, the upregulation of IL-1β and NLRP3, and the activation of caspase-1 and GSDMD were observed in microglia under high glucose exposure [77]. Because caspase-1 and NLRP3 inhibitors prevent microglial cell death, pyroptosis is associated with microglial cell death in DR [77]. A recent study indicated that scutellarin protected RGC pyroptosis in DR via the inhibition of caspase-1, GSDMD, NLRP3, IL-1β, and IL-18 [78]. Collectively, these results suggest that pyroptosis is associated with neurovascular cell death in DR. However, it is not known why these two steps are involved in the plasma membrane rupture during pyroptosis. One possible reason is that the first step (i.e., GSDMD pore formation) may be still a reversible change, and NINJ1-mediated plasma membrane rupture may be a “point of no return”. Thus, pyroptosis may stop before the NINJ1-mediated plasma membrane rupture. Further studies are required to elucidate the association between pyroptosis and the pathogenesis of DR and to establish therapeutic strategies to protect against pyroptosis before the point of no return.

### 2.3. Ferroptosis in DR

Ferroptosis was first reported by Dixon et al. in 2012 as an iron-dependent form of RCD [79]. During ferroptosis, excessive peroxidation of polyunsaturated fatty acids (PUFAs) occurs in the plasma membrane, resulting in the disruption of plasma membrane integrity and cell swelling, such as necrotic cell death [79]. The NCCD defines ferroptosis as RCD initiated by oxidative perturbations of the intracellular microenvironment, constitutively controlled by glutathione peroxidase 4 (GPX4). Ferroptosis is inhibited by iron chelators and lipophilic antioxidants [57]. Ferroptosis does not require caspase activation; therefore, it is thought to be an evolutionarily more classical form of RCD than apoptosis [80]. Although the precise mechanism of ferroptosis remains unclear, two transcription factors, nuclear factor-erythroid 2-related factor 2 (NRF2) and BTB and CNC homology 1 (BACH1), competitively regulate ferroptosis [81]. In addition, three ferroptosis regulatory systems inhibit lipid peroxidation: the glutathione (GSH)–glutathione peroxidase 4 (GPX4), ferroptosis suppressor protein 1 (FSP1), coenzyme Q_10_ (CoQ_10_), and GTP cyclohydrolase 1 (GCH1)–tetrahydrobiopterin (BH_4_) pathways [82]. NRF2 and BACH1 regulate gene expression involving regulatory systems, such as the subunit of system Xc, SLC7A11, FSP1, GCH1, ferritin, and GPX4 [81]. The hypothetical molecular pathways involved in ferroptosis are shown in Figure 3. The Fenton reaction is a chemical reaction which forms toxic hydroxyl radicals (HO•) by reducing H_2_O_2_ in the presence of Fe^2+^ (H_2_O_2_ + Fe^2+^→HO•+ OH^−^ + Fe^3+^) (Figure 3). Because Fenton reactions induce lipid peroxidation, they play a key role in ferroptosis. The GCH1–BH_4_ pathway inhibits phospholipid hydroperoxide (PLOOH), while the GSH–GPX4 pathway catalyzes the reduction of PLOOH (Figure 3).

Growing evidence indicates that ferroptosis is associated with the pathogenesis of diabetes mellitus and its complications, including DR [85]. Ferrostatin-1 is a synthetic compound that acts as a classical hydroperoxyl radical scavenger; however, Miotto et al. indicated that ferrostatin-1 eliminates lipid hydroperoxides and produces the same anti-ferroptotic effect as GPX4 in the presence of reduced iron [86]. Shao et al. indicated that ferrostatin-1 reduces ferroptosis by improving the antioxidant capacity of the Xc-GPX4 pathway in retinal epithelial cell line cultures exposed to high-glucose media and in animal models of DR [87]. Fatty acid binding protein 4 (FABP4) is an independent prognostic marker of DR [88,89]. Fan et al. indicated that FABP4 inhibition alleviates lipid metabolism and oxidative stress by regulating peroxisome proliferator-activated receptor γ (PPARγ)-mediated ferroptosis and reduces ferroptosis by upregulating PPARγ activity in ARPE-19 cells cultured in high-glucose media [90]. In addition, the study suggests that FABP4 inhibition reduces ferroptosis in retinal tissues in a diabetic animal model [90]. Liu et al. indicated that glial maturation factor Β, a neurodegenerative factor that is upregulated in the vitreous in the early stage of DR, is involved in the lysosomal degradation process in autophagy, resulting in ASCL4 accumulation and ferroptosis in RPE cells cultured in high-glucose media [91]. In addition, the study suggests that the ferroptosis inhibitor liproxstatin-1 is effective in protecting retinal tissues in early DR and maintaining visual function in a diabetic rat model in vivo [91]. Liu et al. demonstrated that in human retinal endothelial cells cultured under high-glucose conditions, long non-coding RNA zinc finger antisense 1 (ZFAS1) is upregulated and activates ferroptosis by modulating the expression of ACSL4 [92]. A recent clinical study indicated that compared to those of the normal group, the serum levels of GPX4 and GSH were significantly lower and lipid peroxide, iron, and ROS were significantly higher in patients with DR [93]. Thus, ferroptosis-related biomarkers may be involved in the pathological processes of DR [93]. Natural compounds may effectively inhibit ferroptosis in patients with DR. A recent study indicated that amygdalin, an effective component of bitter almonds, inhibits ferroptosis in human retinal endothelial cells exposed to high glucose levels by activating the NRF2/antioxidant response element signaling pathway [94]. Another study indicates that 1,8-cineole, the main component of volatile oils in aromatic plants, inhibits the ferroptosis of the retinal pigment epithelium under diabetic conditions via the PPARγ/thioredoxin-interacting protein pathways [95]. Although the point of no return of ferroptosis remains unclear, ferroptosis may be a therapeutic target for preventing the progression of DR. Ferroptosis is likely related to vascular cell death in DR. Further studies are required to elucidate the precise mechanisms underlying ferroptosis in the neurovascular impairment in DR.

### 2.4. Necroptosis in DR

The NCCD defines necroptosis as a type of RCD triggered by perturbations of intracellular or extracellular homeostasis which critically depends on the kinase activities of mixed-lineage kinase ligand (MLKL), receptor-interacting protein kinase 3 (RIPK3), and RIPK1 [57]. However, studies on the association between RIPK1 expression and necroptosis are relatively limited [57]. Necroptosis is characterized by a necrosis-like appearance, including cell swelling, mitochondrial membrane permeabilization, and membrane rupture, resulting in an inflammatory reaction in a caspase-independent manner [96]. There are three necroptosis inducers: (1) death ligands which bind with death receptors including TNF-α, Fas, or TNF-related apoptosis-inducing ligand (TRAIL); (2) pathogens which are recognized by TLR family members, such as TLR3 or TLR4; and (3) Z-DNA which is recognized by Z-DNA binding protein 1 (ZBP1) [97]. All intracellular signals from these inducers aggregate into RIPK3. Toll/IL-1R domain-containing adaptor-inducing interferon β (TRIF)-mediated necroptosis and ZBP1-mediated necroptosis are RIPK1 independent [57]. A hypothetical scheme of the molecular pathways involved in necroptosis is shown in Figure 4. RIPK1 was first identified as a regulatory factor in necroptosis [98], and RIPK1 is thought to bind to RIPK3 via self-phosphorylation [99]. However, TRIF and ZBP1 directly bind to RIPK3 and induce necroptosis in an RIPK1 independent manner, and RIPK1 inhibits necroptosis mediated by TRIF and ZBP1 [100,101]. The precise mechanisms of MLKL pores in the plasma membrane are debatable. However, the four-helix bundle (4HB) domain exists at the N-terminus of MLKL, and the 4HB domain is integrated with the plasma membrane and thought to form the MLKL pore [102]. Unlike pyroptosis, NINJ1 does not require membrane rupturing during necroptosis [60]. Therefore, the MLKL pores are completely different from the GSDMD pores. In addition, it is unclear how necroptosis is induced under pathological conditions in vivo. Further studies are required to elucidate the association between necroptosis and pathological events.

Very few studies have demonstrated an association between necroptosis and DR because the mechanisms underlying the induction of necroptosis in vivo remain unclear. A recent in vitro study indicated that in RGCs cultured in high-glucose conditions, the expression of RIPK1 and RIPK3 was significantly increased, and necrostatin-1 protected against retinal ganglion cell necroptosis [103]. Xu et al. indicated that an intravitreal injection of Dickkopf-1 protected streptozotocin-induced diabetic rats against retinal tissue necroptosis in vivo [104]. A recent study indicated that in the diabetic retina, the expression of RIPK1, RIPK3, and MLKL is increased in activated microglia, and that the necroptosis inhibitor GSK-872 reduces neuroinflammation and neurodegeneration, followed by an improvement of visual function in diabetic mice [105]. They concluded that microglial necroptosis was a therapeutic target in early DR [105].

Researchers should be aware that MLKL pore formation is not always a point of no return in the process of necroptosis. Due to the repair mechanisms of the cell membrane, some necroptotic cells with MLKL pores remain alive [106]. Living cells release inflammatory cytokines and induce inflammation [106]. Further studies are required to elucidate the role of necroptosis in the pathogenesis of DR.

## 3. Therapeutic Approaches for Retinal Cell Death in DR

Therapeutic approaches for retinal cell death in DR have mainly focused on neuroprotection and vasoprotection because neuronal cell death is an irreversible change and is directly related to visual function in patients with diabetes. Pericyte loss may be the first trigger for vascular abnormalities in DR, and pericyte protection is a potential therapy for preventing the onset of DR. Endothelial cell loss leads to acellular vessels, resulting in the functional loss of vessels, followed by vascular leakage and capillary occlusion in DR. Therefore, the protection of endothelial cells is a potential therapeutic approach for early DR. In glial cells, the main therapies may regulate overactivation, followed by the amelioration of inflammatory reactions in DR. Thus, this section focuses on neuroprotection and vasoprotection in DR.

A recent study indicated increased levels of sortilin in human diabetic retinas and that sortilin is highly colocalized with the p75 neurotrophin receptor in Müller cells in diabetic retinas [107]. An intravitreal injection of anti-sortilin antibodies had a protective effect on inner retinal cells and RGCs in diabetic mice [107]. The study concluded that sortilin is a novel pharmacological target for the prevention of neurodegeneration in early DR [107]. The in vivo RGC count is one of the most reliable methods to examine neuroprotection because it has protective effects against all types of cell death, apoptosis, necrosis, ferroptosis, and pyroptosis. Huperzine A is a natural alkanoid that is isolated from *Huperzia serrata*. Zhang et al. indicated that huperzine A has a protective effect on diabetic retinas in a diabetic rat model via the phosphorylation of heat shock protein 27 and the activation of the anti-apoptosis signaling pathway [108]. Natural compounds are ideal for preventing early diabetic retinopathy owing to their safety profile. Similarly, a recent study indicated that the oral administration of *Euterpe oleracea* Mart.-enriched foods prevented the reduction in the amplitudes of full-field electroretinograms (ERGs) of diabetic mice [109]. We will attempt to examine these natural compounds in future clinical trials. Memantine, an antagonist of N-methyl-d-aspartate receptors, is used as an anti-Alzheimer’s disease drug. Elsayed et al. demonstrated that oral memantine protects retinal tissues in diabetic mice by suppressing the ROS/thiodoxin-interacting protein/NLRP3 signaling cascade [110]. As memantine suppresses the NLRP3 signaling cascade, it inhibits pyroptosis [111]. Memantine suppresses macrophage pyroptosis in acute lung injury, and the US Food and Drug Administration has approved memantine for the treatment of acute lung injury [111]. However, oral memantine treatment in patients with glaucoma failed to prevent glaucomatous progression in a previous clinical trial [112]. Further randomized clinical trials of memantine for DR are required to elucidate its protective effect. NOX4 is an NADPH oxidase that generates ROS and is involved in DR. Dionysopoulou et al. indicated that the topical administration of the NOX4 inhibitor GLX7013114 reduced VEGF, the activation of caspase-3, and proinflammatory cytokines in diabetic animal models [113]. In addition, pattern ERGs showed that the RGC function was protected with the topical administration of GLX7013114 [113]. As GLX7013114 reduces vascular leakage and protects RGC function, GLX7013114 has neuroprotective, anti-inflammatory, and vasoprotective properties [113]. Topical administration is ideal for treating early DR. Thus, GLX7013114 could be used in future clinical trials. Similarly, the topical administration of cannabinoid receptor 1 antagonists and cannabinoid receptor 2 agonists protected RGC axons and reduced vascular permeability by attenuating nitrative stress in early-stage DR in diabetic rats [114]. Thus, cannabinoid drugs may have both neuroprotective and vasoprotective effects in DR. VEGFA is known to exert a neuroprotective effect via the VEGF receptor 2. The topical administration of nerve growth factor (NGF) reduces inflammatory and pro-apoptotic intracellular signals and maintains VEGF receptor 2 expression in the RGCs of diabetic rats [115]. In the retinas of diabetic rats, the expression of VEGF receptor 2 is decreased. Thus, NGF may contribute to the switch from proangiogenic and apoptotic phases to the neuroprotective phase of VEGF in early DR [115]. Another study indicated that the topical administration of NGF successfully prevented RGC loss, pericyte loss, and acellular capillary development in animal diabetic models [116]. A randomized controlled study on the neuroprotective effects of topical recombinant human NGF in patients with glaucoma has already started [117]. Thus, topical NGF therapy for early DR could be translated into clinical practice in the future. Fang et al. indicate that berberine, a GABA-alpha receptor agonist, reduces RGC apoptosis and improves visual function via the upregulation of protein kinase C-α and Bcl-2 [118]. Jung et al. indicated that orally administered nicotinamide attenuates retinal apoptosis by reducing oxidative DNA damage and supporting DNA repair [119]. Growth differentiation factor 11 (GDP11), which is involved in the regulation of retinal progenitor cells, reduces pericyte loss and retinal microvascular endothelial cell apoptosis in experimental DR via the activation of TGF-β/Smad family member 2 and phosphatidylinositol-3/Akt/forkhead box protein O1 and the inhibition of NF-κB [120]. Thus, GDP11 may be a therapeutic option for vasoprotection in DR. Ginsenoside Rd is an active ingredient isolated from Panax notoginseng and Panax ginseng. Tang et al. suggested that ginsenoside Rd reduces high-glucose-induced endothelial cell apoptosis via the AMP-activated protein kinase–sirtuin-1 interaction [121]. Thus, ginsenoside Rd may be a potential vascular protective drug against early DR. Most therapeutic approaches for retinal cell death in DR seem to target apoptosis, probably because apoptosis is a major form of cell death in DR. Apoptosis is an evolutionarily newer form of cell death than other regulated necrotic cell deaths, and living organisms may preferably select apoptosis because the induction of inflammation followed by the exacerbation of pathological changes can be avoided.

Several studies have suggested the potential of anti-pyroptotic therapies for DR. As mentioned previously, scutellarin, a flavonoid extracted from traditional Chinese medicines, protects against RGC pyroptosis in DR in vivo by inhibiting pyroptosis-related factors [78]. Li et al. demonstrated that high glucose induces pyroptosis rather than apoptosis in human retinal microvascular endothelial cells and that miR-200c-3p attenuates human microvascular endothelial cell pyroptosis by targeting SLC30A7 [122]. Ma et al. indicate that microRNA-192 reduces high-glucose-induced retinal pigment epithelium cell pyroptosis by regulating the FTO-ketoglutarate-dependent dioxygenase/NLRP3 signaling pathway [123]. Although diabetic stress can induce pyroptosis in retinal cells, it remains debatable whether pyroptosis is a major cause of cell death in DR. However, there is no doubt that pyroptosis-related factors, such as NLRP3, caspase-1, IL-1β, and ASC, can be therapeutic targets for DR because DR is a chronic inflammatory disease and pyroptosis is strongly related to inflammation [58].

Although ferroptosis could be a therapeutic target for DR, as described previously [85,87,88,89,90,91,92,93,94,95], few studies have demonstrated the association between necroptosis and the pathogenesis of DR in vivo. In most cases, necroptosis is experimentally induced in vitro. However, a few studies have demonstrated pathologically induced necroptosis in vivo. Further in vivo studies are required to elucidate the association between necroptosis and retinal cell death in DR.

## 4. Conclusions

Until now, we have mainly focused on apoptosis and anti-apoptotic therapies for DR because we have believed that apoptosis is the major cell death form in DR [5,6,12,13,28,29,30,31,36,40,44,45,61]. Recently, however, other forms of RCD, including pyroptosis, ferroptosis, and necroptosis, have been gradually reported in DR. Thus, the main purpose of this review is to introduce these regulated cell death mechanisms in detail. Retinal cell death can be fatal during the development and progression of DR. Retinal neuronal cell death is an irreversible change directly related to vision loss in patients with diabetes. Pericyte loss can trigger vascular abnormalities in DR. Although apoptosis is silent cell death and no inflammatory reaction is induced, RCD including ferroptosis, pyroptosis, and necroptosis induces inflammation in retinal tissues. Because DR is a chronic inflammatory disease, RCD may be involved in retinal cell death in DR. To establish neuroprotective and vasoprotective therapies for DR, multiple comprehensive approaches for preventing multiple forms of cell death should be considered.

## Figures and Tables

**Figure 1 ijms-24-12919-f001:**
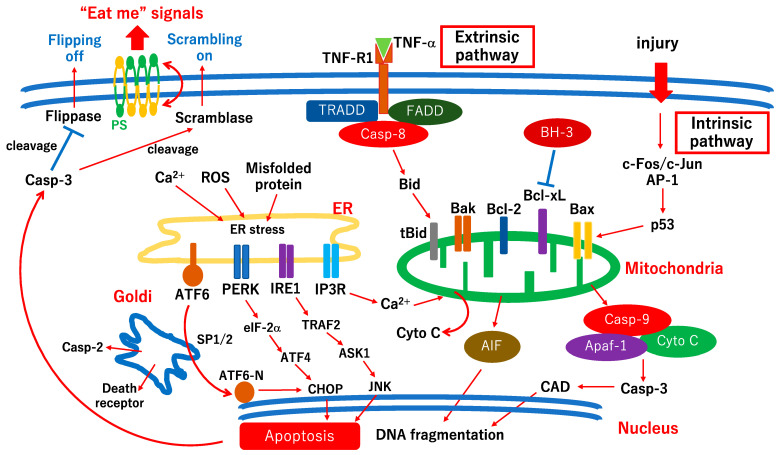
Hypothetic scheme of apoptotic cell death pathways in retinal cells under diabetic stress. The figure is modified and updated from the figure in a previous review [5]. Chronic hyperglycemia is a biochemical injury for retinal cells. Because the cell death mechanisms of the biochemical injury are, in part, common with those of physical injuries, the scheme is mainly made from findings in optic nerve injuries and retinal culture studies [5]. In most neuronal cells, extrinsic pathway signals are thought to be transferred to mitochondria via truncated Bid, probably because most apoptosis-related factors, including pro-caspase-3, may be preserved in the mitochondria in physiological conditions. Thus, regarding neuronal apoptosis, the role of mitochondria may be more critical than that of other cells. In apoptotic cell death, DNA fragmentation is the “point of no return”. AP-1, activator protein-1; Casp-9, caspase-9; Cyto C, cytochrome c; Apaf-1, apoptosis protease-activating factor 1; Bcl-2, B-cell lymphoma 2; Bcl-xL, B-cell lymphoma-extra-large; tBid, truncated Bid; AIF, apoptosis-inducing factor; Casp-3, caspase-3; Casp-8, caspase-8; TRADD, TNF receptor 1-associated death domain protein; FADD, Fas-associated death domain; TNF-α, tumor necrosis factor-α; TNF-R1, tumor necrosis factor receptor 1; ER, endoplasmic reticulum; ROS, reactive oxygen species; IP3R, inositol trisphosphate receptor; IRE1, inositol-requiring ER-to-nucleus signaling protein 1; PERK, protein kinase-like ER eukaryotic initiation factor-2alpha kinase; ATF6, activating transcription factor-6; TRAF2, TNFR-associated factor 2; JNK, c-Jun-N-terminal protein kinase; ASK1, apoptosis signal-regulating kinase 1; eIF-2α, eukaryotic initiation factor-2α; ATF4, activating transcription factor-4; CHOP, CCAAT/enhancer-binding protein homologous protein; Casp-2, caspase-2; PS, phosphatidylserine. SP1/2, site-1/2 protease.

**Figure 2 ijms-24-12919-f002:**
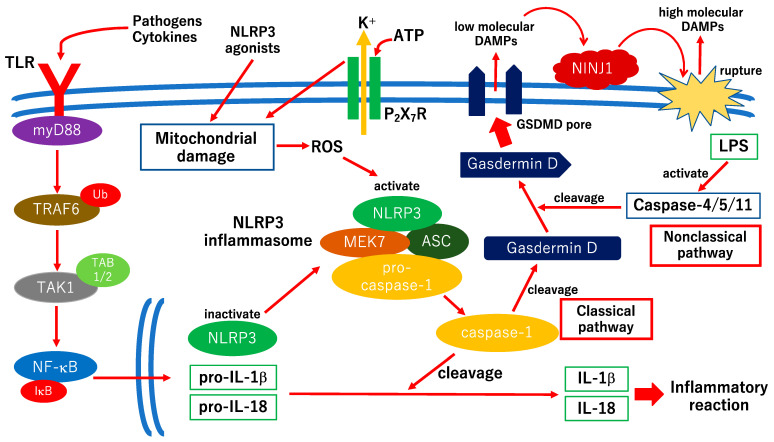
Hypothetical scheme of molecular mechanisms of pyroptosis. The Nomenclature Committee on Cell Death does not recommend the use of alternative terms such as “pyronecrosis” [57]. The central molecule of pyroptosis is gasdermin D (GSDMD). There are two pathways which activate GSDMD, the classical pathway and the non-classical pathway. In the classical pathway, Nod-like receptor family pyrin domain containing 3 (NLRP3) inflammasome is activated under the Toll-like receptor (TLR) signals which results in caspase-1 activation. Caspase-1 cleaves pro-interleukin-1β (pro-IL-1β) and pro-IL-18 in addition to GSDMD. The N-terminal domains of GSDMDs form plasma membrane pores and release low-molecular-damage-associated molecular patterns (DAMPs). On the other hand, in the non-classical pathway, caspase-4/5/11 are activated by endotoxins, such as lipopolysaccharide (LPS). Activated caspase-4/5/11 can cleave GSDMD followed by the formation of GSDMD pores. Further inflammatory signals activate nerve injury-induced protein 1 (NINJ1). Cell-surface NINJ1 mediates further plasma membrane ruptures during pyroptosis followed by releasing high-molecular-weight DMAPs [60]. Thus, pyroptosis can passively exacerbate inflammatory reaction via NINJ1-mediated plasma membrane rupture. MyD88, myeloid differentiation primary response gene 88; TRAF6, TNF receptor-associated factor 6; Ub, ubiquitin; TAK1, TGF-β-activated kinase 1; TAB1/2, TAK1-binding protein 1/2; IκB, inhibitor of κB; NF-κB, nuclear factor-κB; ROS, reactive oxygen species; MEK7, MAPK ERK kinase 7; ASC, apoptosis-associated speck-like protein containing a CARD; P_2_X_7_R, P_2_X_7_ receptor.

**Figure 3 ijms-24-12919-f003:**
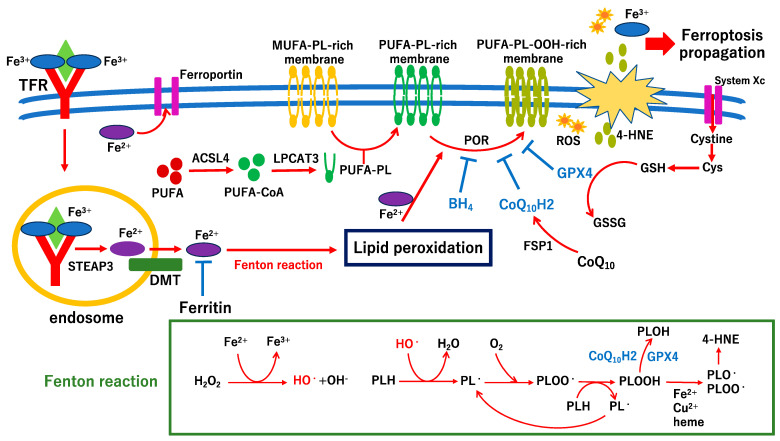
Hypothetical scheme of molecular pathways of ferroptosis and Fenton reaction. The late phase of ferroptosis is still considered a “black box”, and thus, it is unclear where the “point of no return” is. However, before membrane integrity disruption, propagation of ferroptosis occurs [83] probably because lipid peroxidation-associated factors, including iron and heme, may be distributed to surrounding cells before the membrane ruptures [84]. The affected points of endogenous inhibitory factors, GPX4 and CoQ_10_H2, are shown in the figure. The scheme of the Fenton reaction is shown in the green box. Acyl-CoA synthetase long-chain family member 4 (ACSL4) and lysophosphatidylchoiline acyltransferase 3 (LPCAT3) are major regulatory enzymes of ferroptosis [80,82]. ASCL4 catalyzes to connect acyl-CoA with a polyunsaturated fatty acid (PUFA; PUFA-CoA). LPCAT3 catalyzes the translocation of PUFA-CoA into acyl phospholipids which results in synthesis of PUFA-phospholipid (PL)-rich membrane. The PUFA-PL-rich membrane increases the sensitivity of lipid peroxidation followed by facilitation of ferroptosis. NADPH-cytochrome P-450 reductase (POR) peroxidates PUFA-PL (PUFA-PL-OOH) as an electron donor of nicotinamide adenine dinucleotide phosphate (NADPH). Transcription factors NRF2 and BACH1 may control POR via regulating transcription of NADPH quinone dehydrogenase 1, which results in facilitating or inhibiting ferroptosis. TFR, transferrin receptor; STEAP3, six-transmembrane epithelial antigen of the prostate 3; DMT, divalent metal transporter; GSH, glutathione synthase; GSSG, oxidized glutathione; Cys, cysteine; 4-HNE, 4-hydroxynonenal; CoQ_10_H2, reduced coenzyme q10; GPX4, glutathione peroxidase 4; BH_4_, tetrahydrobiopterin; FSP1, ferroptosis suppressor protein 1; HO•, hydroxy radical; PL•, PLOO•, PLO•, phospholipid hydroxy radical; PLOOH, phospholipid hydroperoxide.

**Figure 4 ijms-24-12919-f004:**
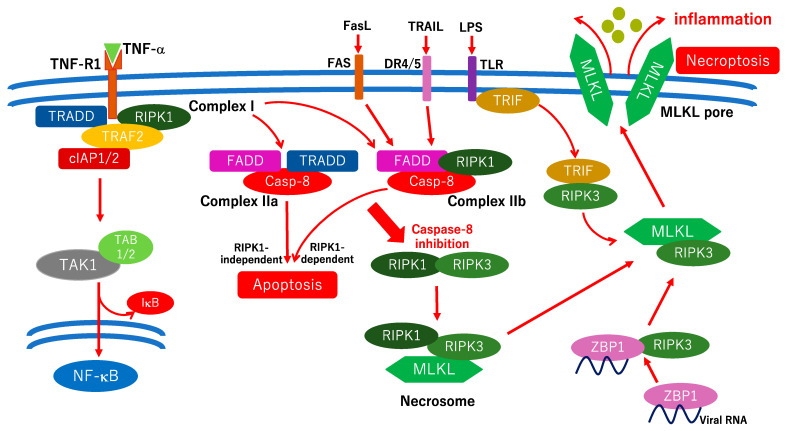
Hypothetical scheme of the molecular pathways of necroptosis. Binding TNF-α with TNF-R1 induces complex I formation which includes TRADD, TRAF2, RIPK1, and cellular inhibitors of apoptosis protein 1/2 (cIAP1/2). Once NF-κB target protein synthesis is inhibited, complex IIa is activated followed by caspase-3 activation and induction of apoptosis in an RIPK1-independent manner. The inhibition of RIPK1 ubiquitination or phosphorylation induces complex IIb activation which results in RIPK1-dependent apoptosis. Once caspase-8 is inhibited, RIPK1, RIPK3, and MLKL form the necrosome which results in MLKL phosphorylation and oligomerization. MLKL integrates with the plasma membrane and forms the MLKL pore. TRIF- and ZBP1-mediated necroptosis are independent of RIPK1. RIPK3 is a more critical factor than RIPK1 in the process of necroptosis. FasL, Fas ligand; DR4/5, death receptor 4/5.

## Data Availability

Not applicable.

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
