# Peer review of "Neurovascular Cell Death and Therapeutic Strategies for Diabetic Retinopathy"

_ijms, 2023, doi:10.3390/ijms241612919_

Round 1

Reviewer 1 Report

Comprehensive and detailed review about the neuronal cell death mechanisms in diabetic retinopathy, namely apoptosis, ferroptosis, necroptosis and pyroptosis. The information is accurate, up-to-date and easy to read. The schemes are very illustrative. I do not have any negative comments but a minor concern. The author self-citate himself a relatively high amount of times (14/123). Despite this, most of his papers (8/14 or more) are research papers, so I find it acceptable.

Any criticism about adding more information in this, I would find it reduncdant or out of the scopus of this review.

Author Response

Thank you for your excellent review comments.

Reviewer 2 Report

The author published five reviews on the same subject over the last 3 years. Of them, “The Pathogenesis and Therapeutic Approaches of Diabetic Neuropathy in the Retina” recently published in IJMS appears quite similar to that entitled “Neurovascular cell death and therapeutic strategies for diabetic retinopathy” to be revised here for the same journal. The author needs to emphasize the added value of the present review in respect to previous papers by either  himself  or others on the same subject.  The concept of DR as a neurovascular disease is well established. In this respect, neuroprotecting therapies have been largely described as in the present review. Here, neuroprotectants should be listed according to a logical thread instead of a mere catalogue of potential therapies and their added value should be highlighted.  If an author has a well established reputation as in this case, then his main responsability would be to shed some light on his argument of interest by focussing it from different points of view to avoid repetitions on the same subject. 

English language does not require major revision

Author Response

Until now, we have mainly focused on apoptosis and anti-apoptotic therapies for DR because we have believed that apoptosis is the major cell death form in DR [5,6,12,13,28-31,36,40,44,45,61]. Recently, however, other regulated cell death forms including pyroptosis, ferroptosis, and necroptosis have been gradually reported in DR. Thus, the main purpose of this review is to introduce these regulated cell death mechanisms in detail and thus, therapeutic strategies are optionally described. Because most therapies targeted these regulated cell death forms seem to be still far from clinical practice. Instead of that, all figures of these cell death pathways have been updated and prepared accurately as much as possible. For example, in most reviews, the cell death pathways of pyroptosis have been finished in the formation of the GSDMD pore. But it is not correct or readers may misunderstand the GSDMD pore is the final step of pyroptosis. In this review, we described the NINJ1-mediated plasma membrane rupture as the final step. In case of ferroptosis, in addition to Fenton reaction, ferroptosis propagation are also included in the text and the figure. In case of necroptosis, some reviews describe the TRIF-RIPK3-mediated pathway is not described in a RIPK1 independent manner. Because it is not accurate, we have revised it in the figure correctly. In case of apoptosis, most reviews do not include “Eat me” signals mediated by flippase and scramblase. Thus, we included the “Eat me” signal as the final step in apoptosis. Although it is important to list up neuroprotectants for DR, this is out of the scope of this review as described by reviewer 1. Instead of that, therapeutic options are distributed in the text as giving clues to establish the therapeutic options for DR for researches. We have added some comments described above in the conclusion.

Reviewer 3 Report

This is a seminal narrative review of retinal neurovascular
cell death under diabetic conditions. It is an important complication of T2D.

The author describes
apoptosis, pyroptosis, ferroptosis, and necroptosis, covering
all the possible death pathways, as well as the therapeutic
approaches that could be applied to prevent these detrimental
events for the retina. This review can be used as a starting
point to look for experimental ideas and design research
protocols. 

The references are appropriate.

I enjoyed the figures which accurately describe the molecular pathways of cell death described in the manuscript. In summary it is an excellent narrative review of the underlying molecular mechanisms and possible interventions applied in a very important clinical complication of T2D. This review is perfectly matching to IJMS.

Author Response

We have appreciated your highest evaluation for our manuscript.